# Optimal Control Strategy for Parallel Plug-in Hybrid Electric Vehicles Based on Dynamic Programming

**Ying Tian, Jiaqi Liu \*, Qiangqiang Yao and Kai Liu**

Beijing Key Laboratory of Powertrain for New Energy Vehicle, School of Mechanical, Electronic and Control Engineering, Beijing Jiaotong University, Beijing 100044, China; ytian1@bjtu.edu.cn (Y.T.); 18116027@bjtu.edu.cn (Q.Y.); 16125954@bjtu.edu.cn (K.L.)
\* Correspondence: 20121269@bjtu.edu.cn; Tel.: +86-010-5168-8408

**Abstract:** In this paper, the dynamic programming algorithm is applied to the control strategy design of parallel hybrid electric vehicles. Based on MATLAB/Simulink software, the key component model and controller model of the parallel hybrid system are established, and an offline simulation platform is built. Based on the platform, the global optimal control strategy based on the dynamic programming algorithm is studied. The torque distribution rules and shifting rules are analyzed, and the optimal control strategy is adopted to design the control strategy, which effectively improves the fuel economy of plug-in hybrid electric vehicles. The fuel consumption rate of this parallel hybrid electric vehicle is based on china city bus cycle (CCBC) condition.

**Keywords:** hybrid powertrain; control strategy; dynamic programming; real-time simulation





## 1. Introduction

In order to solve the increasingly serious problems of energy shortages and environmental pollution, new energy vehicles have received extensive attention and rapid development due to their good economy, low emissions, and long driving range. A plug-in hybrid vehicle is a new energy vehicle that uses an external grid to charge and has two drive systems, an engine and a motor. Hybrid vehicles are characterized by coordinated participation of electric drive system, allowing engine control to operate in high-efficiency, low-pollution areas, improving fuel economy and reducing emission [1]. Therefore, with the current technical background, hybrid electric vehicles are one of the more effective technical routes [2]. The hybrid control unit (HCU) is the control core of hybrid electric vehicles. The control strategy loaded in the HCU is responsible for coordinating the torque distribution of the engine and the motor and controlling transient processes such as shifting and clutch coupling and disengagement to achieve better fuel economy and driving comfort.

In recent years, control algorithms have been continuously optimized, and many new algorithms have appeared. Roman et al. [3] combined active disturbance rejection control with proportional derivative Takagi–Sugeno fuzzy control adjusted by virtual reference feedback, which was used to automatically and optimally adjust the parameters in a model-free manner, and can reduce the calculation time.

At present, the control algorithms of hybrid electric vehicles developed by research institutions and vehicle companies can be divided into three categories, namely rule-based control algorithms, control algorithms based on optimization theory, and intelligent control algorithms. Control algorithms based on optimization theory are divided into real-time optimization algorithms and global optimization algorithms.

The control strategy Peng et al. [4] combined the instantaneous efficiency of the engine, motor, and battery with actual operating conditions, such as engine, motor, battery temperature, and braking energy recovery, to obtain the best combination of engine and motor energy. Then, the optimal operating points of the engine and electric motor were

determined according to the principle of optimal efficiency of the system. Aiming at ISG hybrid electric vehicles, Zhao et al. [5] proposed a hierarchical traction control system based on the multi-objective dynamic coordinated control. Different control strategies have been developed on different levels of controller. Li et al. [6] proposed a fuzzy control strategy that takes the ratio of the required torque to the optimal torque of the engine and the battery state of charge (SOC) as the input and the optimal torque coefficient of the engine as the output. The fuel saving rate was greatly improved. Yang et al. [7] optimized the fuzzy control strategy through the comprehensive constraint conditions of the economic performance and the emission performance of the genetic algorithm to achieve better economy and emission reduction under the typical conditions of NEDC. Gissing et al. [8] integrated fuzzy control and the genetic algorithm into the energy management strategy to optimize the intelligent control algorithm. The genetic algorithm then optimizes the membership function in the control strategy to achieve the requirement of reducing fuel consumption. The rule-based control strategy is completed by algorithm engineers based on the test data and engineering iteration experience. It cannot be changed after loading, so it cannot cope with the parameter drift caused by the fatigue loss of the auto parts. At the same time, due to the constraints of the system's real-time and stability requirements, it is not possible to better reduce fuel consumption and optimize emissions.

Based on heuristic engineering experience, Gino et al. [9] proposed the equivalent consumption minimization strategy (ECMS), which determines the torque demand of the current working condition from the difference between the cycle speed and the actual vehicle speed, and determines the current working mode through the control strategy. The electric energy in the hybrid system is equivalent to fuel consumption in the form of an equivalent factor, and the ECMS algorithm is used to determine the instantaneous optimal energy distribution to determine the minimum equivalent fuel consumption. Zhang et al. [10] used flexible torque request to jointly optimize the torque distribution and shift command based on the ECMS. Li et al. [11] proposed to integrate the fuzzy inference system (FIS) for online reference SOC estimation and the adaptive update law with a flow recognition function into the main framework of the adaptive equivalent power consumption minimization strategy (A-ECMS). Zhang et al. [12] proposed an energy management method based on an approximate Pontryagin's minimum principle (A-PMP), which uses the approximate Hamiltonian function to jointly optimize the torque distribution, shift command, and battery aging performance under the approximate PMP framework. Although the real-time optimization control algorithm ignores the influence of many conditions, such as driving condition and mileage, the control effect of the algorithm is not as good as that of the global optimal algorithm.

Hu et al. [13] used a mapping-based strategy combined with a dynamic programming (DP) algorithm to distribute power between the engine and the battery and determine the minimum fuel consumption. Yang et al. [14] proposed approximate optimization rapid dynamic programming (Rapid-DP) based on the DP method, which effectively reduced the decision-making time. Combined with particle swarm optimization (PSO), it was proved that the multi-mode configuration with the best component parameters is the most fuel-efficient. Harselaar et al. [15] proposed two methods to improve the implementation of DP, with which the calculation time was reduced by up to 66% in the study of three different hybrid powertrain topologies. In order to achieve the best power distribution, Faras et al. [16] developed a dynamic and efficient energy management system that uses a weighted improved dynamic programming algorithm for pre-driving offline optimization and a PID controller for online optimization. In this method, the weight is included in the fitness function, which improves the convergence speed in the long continuous driving cycle. Moura et al. [17] studied the optimal drive power demand allocation problem between different actuators (engine and motor) in a plug-in hybrid vehicle using stochastic dynamic programming to optimize the energy management strategy of the hybrid system. The proportion of fuel and electricity used in the PHEV was clarified, and the impact of relative fuel-electricity price change on optimal PHEV energy management was examined.

In terms of improving fuel economy, the globally optimized energy management strategy can indeed improve the control effect, but it is necessary to know the working conditions of driving vehicles in advance. On the other hand, as the mileage of the working conditions increase, the calculation amount and calculation time of the global optimization method also increase significantly. The processing capability of the current automotive controller processor cannot be used in real-time scenarios.

Wang et al. [18] trained the pattern recognition neural network based on the optimal results of dynamic programming, and used common state correction and relaxation constraints to establish an online model predictive control framework based on speed prediction for online model prediction. Based on the Q-learning algorithm, Xu et al. [19] have carried out a parameterized study on the key factors of the energy management system of the parallel hybrid electric vehicle to realize the torque distribution between the engine and the motor. Lian et al. [20] proposed a method based on transfer learning to realize cross-type knowledge transfer between energy management strategies (EMSs) based on deep reinforcement learning to automatically improve the efficiency of hybrid vehicle energy management strategy development. The intelligent control algorithm is not mature enough and its application in the field of vehicle energy management needs further exploration.

Since the dynamic programming method does not have too many restrictions on the system state equation and performance index function, the system model can be a numerical model based on experimental data. In addition, the fuel consumption problem under the specified path of the hybrid electric vehicle also satisfies the optimization principle of the dynamic programming method and has no aftereffect or overlapping sub-problems. Therefore, the dynamic programming method is more suitable for solving the optimal control problem of the hybrid electric vehicle. The value of the optimal energy management strategy based on dynamic programming is mainly reflected in the following two aspects. First, the dynamic programming algorithm can obtain the optimal result for the hybrid electric vehicle on the specified path, it can be used for control strategies based on rules or other algorithms. Subsequently, effective control parameters can be extracted from the data obtained by the dynamic optimization method to optimize and modify the control strategy of the hybrid electric vehicle.

This paper adopts the model-based controller development process and the research method of V-type development to realize the development of and research on the energy management strategy of parallel plug-in hybrid electric vehicles.

## 2. Parallel Hybrid System Modeling

### 2.1. Powertrain Structure

A schematic diagram of the hybrid power system structure is shown in Figure 1. The research object adopts a single-shaft parallel structure. The output torque of the engine and the motor is transmitted to the wheels by the torque coupling device, the gearbox and the final drive. The real-time communication between the component controller and the vehicle controller is controlled by the vehicle controller.

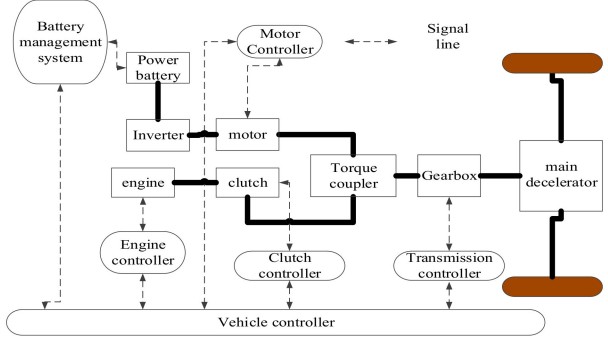

**Figure 1.** Schematic diagram of the hybrid power system structure.

In hybrid electric vehicles, components such as engine, motor, and gearbox have complex dynamic characteristics, and it is difficult to mathematically describe the entire dynamic process. Therefore, the study neglects the secondary factors that affect the system to simplify and facilitate the design of component models and control rules while ensuring that the model can reflect the basic dynamic characteristics of real hybrid electric vehicles. The forward method is used to develop the model in the modeling process and the backward method is used to develop the control strategy based on the dynamic programming algorithm in Section 3. The same model parameters are used in the above two methods.

### 2.2. Key Component Models

2.2.1. Battery Model

The power battery pack is an extremely important energy storage device for plug-in hybrid vehicles. This paper only considers the fuel economy and power of the hybrid system under one cycle condition, the cycle time lasts for a few minutes. The open circuit voltage and the internal resistance (R-int) have large changes, so the impact is ignorable [21]. So, this paper uses an R-int model.

Since a detailed description of the characteristics of the power battery and its influencing factors will make the model extremely complicated, it is generally considered that the open circuit voltage and the R-int of the battery are fitted as functions of SOC by experimental data [22], and the functional relationship is shown in Figure 2. It can be seen from the Figure 2 that when the battery SOC is greater than 10%, the open circuit voltage of the battery changes smoothly, and the maximum fluctuation range is only 4.29%; within the range of SOC from 0% to 100%, the R-int of the battery changes slightly, and the maximum fluctuation range is only 5.08%. The power battery and the R-int of the battery are stable.

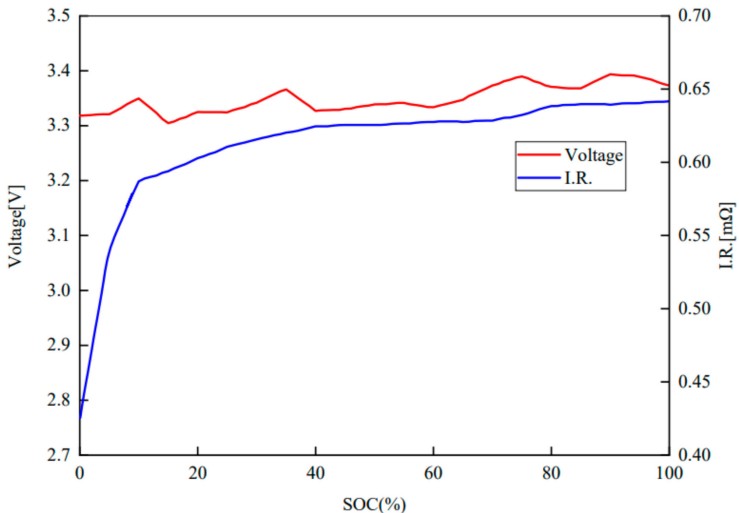

**Figure 2.** Relationship between open circuit voltage/internal resistance and the SOC of a single cell.

The power battery model is mainly established by SOC functions. The current SOC of the power battery is calculated through the initial SOC value, battery capacitance, and circuit current, which is used to obtain the open circuit voltage and resistance value. The input parameter of the whole model is the battery current $I_{bat}$ and the output parameter is the battery voltage $V_{bat\_out}$. The mathematical model of the power battery used in this paper is

$$V_{bat\_out} = V_{bat\_oc} - R_{bat} \cdot I_{bat} \tag{1}$$

$$R_{bat} = f_{batR}(x_{soc}) \tag{2}$$

$$V_{bat\_oc} = f_{bat_{oc}}(x_{soc}) \tag{3}$$

where $V_{bat\_out}$ is the battery terminal voltage; $I_{bat}$ is the battery charge and discharge current; $R_{bat}$ is the battery equivalent internal resistance; and $V_{bat\_oc}$ is the open circuit voltage that varies with the SOC.

The SOC is calculated by using the hour integration algorithm, which is

$$SOC(t) = \frac{C \cdot SOC(t_0) - \int_{t_0}^{t} I_L dt}{C} \tag{4}$$

where $C$ represents the capacity of the electricity, the unit is A·h; and $SOC(t_0)$ is the initial SOC value at time $t_0$.

### 2.2.2. Engine Model

The fuel consumption and torque MAP of the electronically controlled injection diesel engine are obtained by a test bench to describe the external characteristics of the engine. The engine's mathematical model includes current throttle calculation module, torque calculation module and fuel consumption calculation module.

The engine throttle calculation needs to consider three control states of the engine. The first state is the engine idle speed control. In the first state, the PI adjustment calculation is performed according to the current speed and the idle speed. The second state is engine torque control, in which the current engine throttle is equal to the engine throttle command. The third state is the engine speed control, which is obtained by adjusting the current speed and the target speed. After calculating the throttle in the three control states, the actual throttle of the engine takes the three maximum values.

The output torque calculation module calculates the engine output torque by the engine output speed and the engine start command, shown in Figure 3a. The mathematical model of the torque calculation module is

$$T_{eng} = f(\alpha, n) \tag{5}$$

where $\alpha$ is the engine throttle, the range is 0–100%; $n$ is the current engine speed (r/min), and $T_{eng}$ is the actual torque output of the engine (Nm), which is based on the current engine speed and the current throttle checklist.

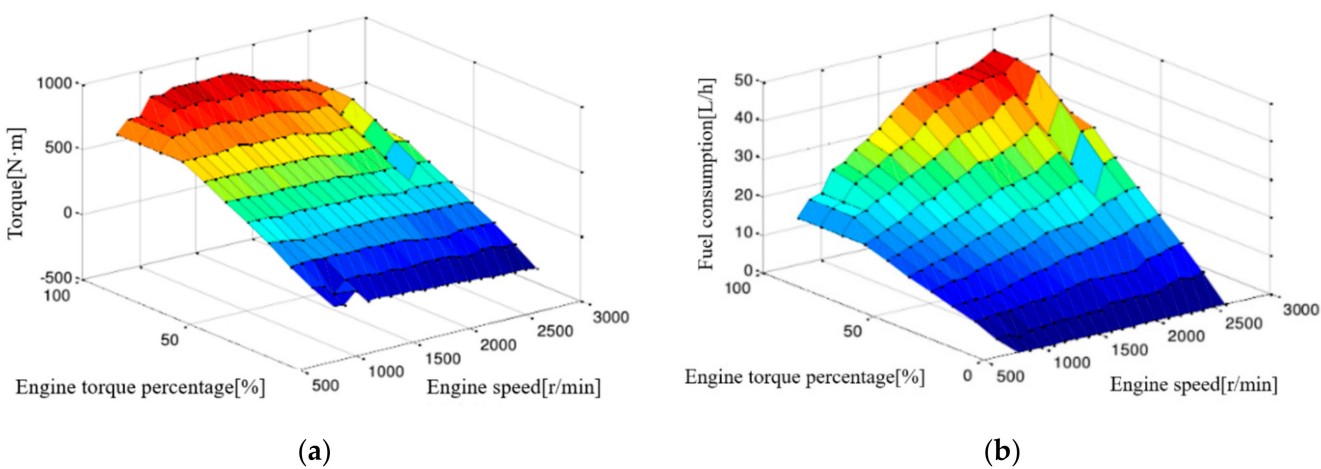

**(a)**　　　　　　　　　　　　　　　　　　　　　　　　　　　**(b)**

**Figure 3.** Engine torque and fuel consumption map. (**a**) Engine torque map. (**b**) Engine fuel consumption map.

The fuel consumption calculation module can obtain the fuel consumption rate by inputting information such as the throttle opening value and the output speed, shown in Figure 3b. The mathematical model of the fuel consumption calculation module is

$$\overset{\bullet}{V}_{fuel} = f(T_{eng}, n) \tag{6}$$

where $\overset{\bullet}{V}_{fuel}$ in units of L/h is the instantaneous fuel consumption based on the current torque and the current speed.

### 2.2.3. Motor Model

The motor mathematical model includes two sub-modules, a current torque calculation module and an output current calculation module. The mathematical model of the current torque calculation module is

$$T_{mot} = \min(T_{cmd}, T_{max}) \tag{7}$$

$$T_{max} = f(V_{bat}, n_{mot}) \tag{8}$$

where $T_{mot}$ is the actual output torque of the motor; $T_{cmd}$ is the torque demand value; $n_{mot}$ is the motor speed; $V_{mot}$ is the motor voltage; and $T_{max}$ is the motor maximum torque, which is obtained by a function of the motor speed and voltage. The functional relationship is shown in Figure 4.

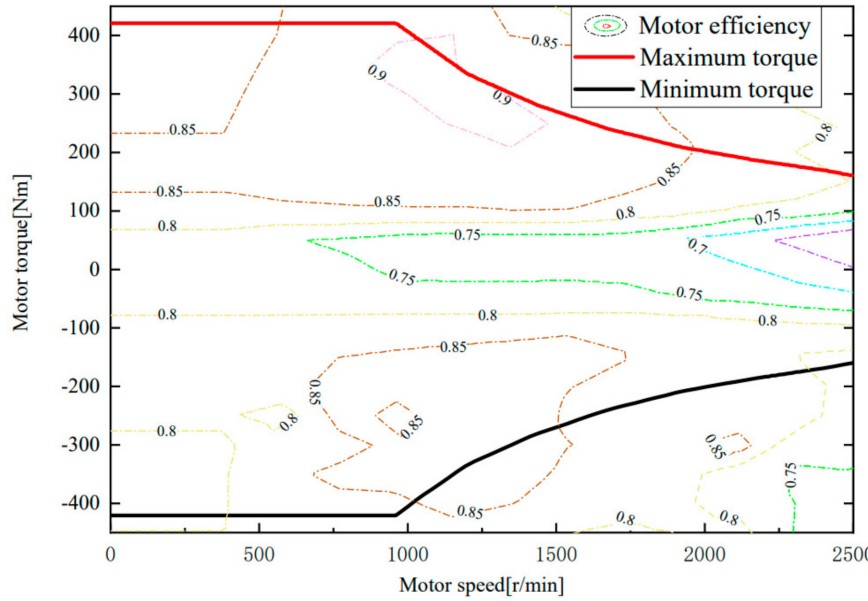

**Figure 4.** Motor efficiency map.

The mathematical model of the output current calculation module is

$$I_{mot} = \begin{cases} P_{mot} \times 1000/(\eta_{mot} \times V_{mot}) & (P > 0) \\ P_{mot} \times 1000\eta_{mot}/V_{mot} & (P \leq 0) \end{cases} \tag{9}$$

$$\eta_{mot} = f(n_{mot}, T_{mot}) \tag{10}$$

where $P_{mot}$ is the motor output power; $\eta_{mot}$ is the motor efficiency; and $I_{mot}$ is the motor current.

### 2.2.4. Transmission Model

Based on the up/down gear rule curve, the gearbox model established for an automated manual transmission (AMT) includes a current gear calculation module, a output torque calculation module, and a moment of inertia calculation module. The three calculation modules calculate the input shaft speed, current gear, output torque, and output

moment of inertia by the input parameters, such as gear command, output speed, input shaft torque, and input moment of inertia. The mathematical model is

$$T_{out} = T_{in} \cdot f(x_{gb}) \cdot 0.95,$$ (11)

$$n_{in} = \begin{cases} n_{in} \cdot f(x_{gb}) & x_{gb} \neq 0 \\ \frac{60}{2\pi} \int \frac{T_{in} - 0.008n_{in\_init}}{T_{in}} + n_{in\_init} & x_{gb} = 0 \end{cases}$$ (12)

where $T_{in}$ is the gearbox input torque (Nm), $T_{out}$ is the gearbox output torque (Nm), $f(x_{gb})$ is a function of the gear command value to obtain the gear ratio, $n_{in}$ is the speed of the gearbox input shaft (r/min), and $n_{in\_init}$ is the gearbox input speed at the beginning of the shift (r/min).

### 2.2.5. Vehicle Model

The vehicle driving resistance includes rolling resistance, ramp resistance, acceleration resistance, wind resistance and other types of resistance. The motor and the engine provide the driving force to overcome the resistance of the vehicle during driving to maintain normal vehicle travel.

The vehicle model in this paper is based on the vehicle travel equilibrium equation. The equilibrium equation is as follows

$$F = mgf \cos \alpha + mg \sin \alpha + \frac{C_D A}{21.15} u^2 + \delta m \frac{du}{dt}$$ (13)

where $F$ is the driving force of the vehicle, m is the mass of the vehicle, g is the acceleration of gravity, $f$ is the rolling resistance coefficient, $\alpha$ is the slope degree, $C_D$ is the drag coefficient, $A$ is the windward area, $u$ is the vehicle speed, and $\delta$ is the rotating mass conversion factor.

### 2.3. Controller Model

The vehicle controller is one of the core components of a hybrid vehicle. Its main function is to coordinate the torque distribution of the engine motor and complete transient control functions such as shifting, clutch disengagement, and engine start and stop, which provides a guarantee that the vehicle will drive normally. The control effect directly affects the vehicle power, economy, comfort, and safety [23].

The hybrid vehicle studied in this paper has three different working modes, namely pure electric, pure engine, and four-wheel hybrid drive. According to the working mode, based on MATLAB/Simulink software, the hybrid design strategy was developed using a modular design method. The controller model of the real-time simulation test is shown in Figure 5.

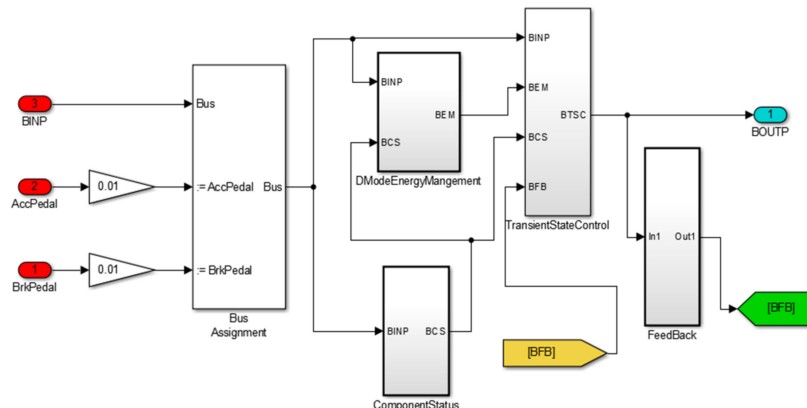

**Figure 5.** Controller model.

As shown in Figure 5, the vehicle controller model has four parts, among which the function modules are a component status judgment module, an energy management module, a dynamic coordination control module, and a feedback module.

## 3. Control Strategy Based on the Dynamic Programming Algorithm

### 3.1. Dynamic Programming Theory

The dynamic programming algorithm divides the process of the problem into several interconnected stages, and appropriately selects state variables, decision variables and optimal value functions, so as to transform a large problem into a family of sub-problems of the same type. The calculation starts from the boundary conditions, recursively to find the optimal solution. In the solution of each sub-problem, the optimization results of the previous sub-problems are used in sequence, and the optimal solution obtained from the last sub-problem is the entire problem.

Figure 6 indicates the energy management multi-stage decision-making process. As shown in Figure 6, in the multi-stage decision-making process, the dynamic programming method is an optimization method that not only separates the current segment from the future segments, but also combines the current benefits with future benefits. When seeking the optimal strategy for the whole problem, for the initial state is known, the decision at one stage is a function of the state at that stage. Therefore, the states of the optimal strategy at each stage can be changed one by one, and the optimal route is determined.

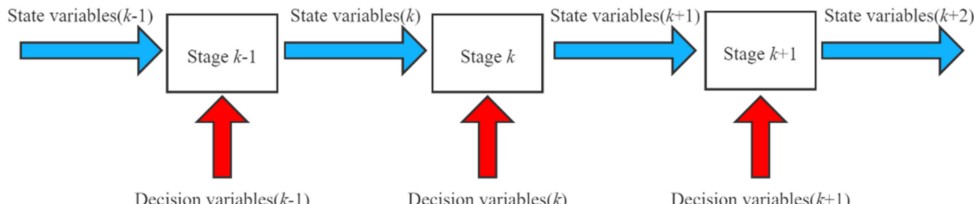

**Figure 6.** Block diagram of the energy management multi-stage decision-making process.

### 3.2. Control Strategy

Knowing the future road conditions, a future trip is divided into several interconnected phases according to time or space, and the driving information of the vehicle at the beginning of each stage is known, such as speed, acceleration, gearbox ratio, battery charging status, etc. Other objective conditions are state variables and phase variables. According to the above three variables, it can be judged whether it is necessary to distribute power between the electric drive system and the engine in order to achieve the goal of reducing fuel consumption and emissions. The energy management problem is actually a multi-stage decision problem with a chain structure [24]. It is feasible to establish an optimization model with the lowest fuel consumption as the goal, and apply the above dynamic programming algorithm to solve the optimization results of the optimal control strategy. The energy management strategy calculation process based on dynamic programming is shown in Figure 7.

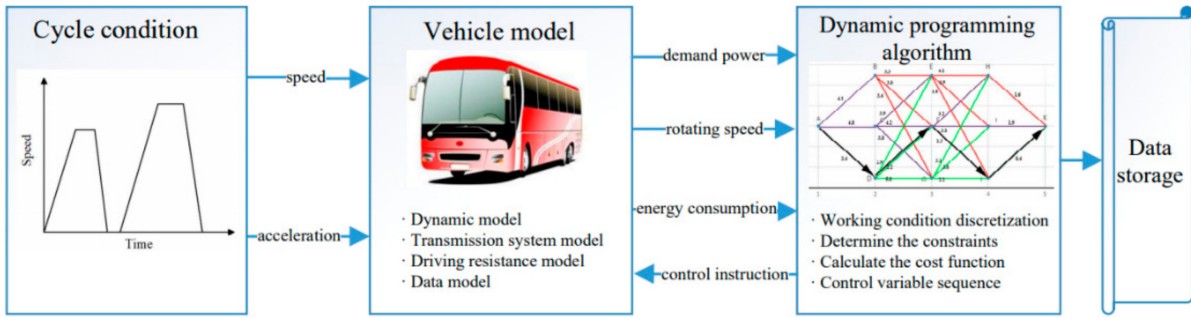

**Figure 7.** Energy management strategy calculation process based on dynamic programming.

The CCBC condition is shown in the Figure 8 as the operating condition for simulation. In order to meet the speed and torque requirements of the vehicle during the driving process, main components of the hybrid electric vehicle must first meet the following inequality constraints before the torque distribution calculation can be performed to obtain a safe and stable driving experience.

$$
\begin{cases}
SOC_{min} \leq SOC(k) \leq SOC_{max} \\
\omega_{eng\_min} \leq \omega_{eng}(k) \leq \omega_{eng\_max} \\
T_{eng\_min} \leq T_{eng}(k) \leq T_{eng\_max} \\
\omega_{mot\_min} \leq \omega_{mot}(k) \leq \omega_{mot\_max} \\
T_{mot\_min} \leq T_{mot}(k) \leq T_{mot\_max}
\end{cases}
\tag{14}
$$

where $SOC_{min}$ and $SOC_{max}$ are the minimum and maximum battery state of charge, respectively, $\omega_{eng\_min}$ and $\omega_{eng\_max}$ the minimum and maximum engine speed (r/min), respectively, $\omega_{eng}(k)$ is the engine speed at time k, $T_{eng\_min}$ and $T_{eng\_max}$ are the minimum and maximum engine torque at the speed of $\omega_{eng}(k)$, respectively, $T_{eng}(k)$ is the actual torque of the engine at the speed of $\omega_{eng}(k)$, $\omega_{mot\_min}$ and $\omega_{mot\_max}$ are the minimum and maximum motor speed, respectively, $\omega_{mot}(k)$ is the motor speed at time k, $T_{mot\_min}$ and $T_{mot\_max}$ are the minimum and maximum motor torque at the speed of $\omega_{mot}(k)$, respectively, and $T_{mot}(k)$ is the actual torque of the motor at the speed of $\omega_{mot}(k)$.

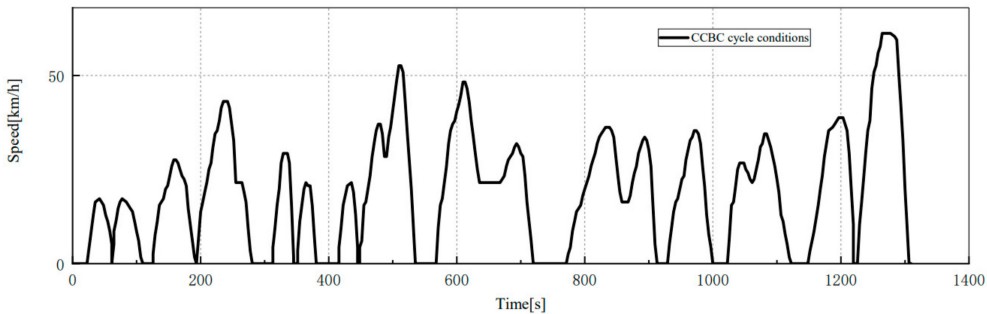

**Figure 8.** CCBC condition.

The optimization problem of energy management for hybrid electric vehicles generally takes the comprehensive fuel consumption as the objective function. According to the vehicle speed demand, the torque demand of the power components, and the gear shift demand at different moments, different state variables will be generated. Moreover, changes in state variables at each moment will also have a significant impact on the fuel consumption. Since the battery SOC is a state variable that changes with the passage of time, and the SOC has no direct relationship between the previous time period and the next time period, the SOC is selected as the state variable, and the engine torque and gear shift action are regarded as two decision variables of the dynamic programming algorithm.

Therefore, the state variable $x(k)$ of the $k$th stage can be expressed by the battery SOC as follows:

$$x(k) = \{SOC(k)\}, \ k = 0, 1, 2, \cdots, N \tag{15}$$

The decision variable $u(k)$ of the $k$th stage can be expressed as follows with engine torque $T_{eng}(k)$ and shift action $shift(k)$:

$$u(k) = \{T_{eng}(k), shift(k)\}, \ k = 0, 1, 2, \cdots, N \tag{16}$$

The optimization object of this paper is the fuel consumption of the vehicle, so the comprehensive fuel consumption rate of the hybrid electric vehicle is taken as the optimization target of the energy management strategy. Therefore, the system's cumulative cost optimization objective function J is the sum of the cost objective functions of each stage, and the stage cost objective function L is defined as the equivalent fuel consumption of the $k$th stage, which is composed of the engine's fuel consumption and the equivalent electrical energy consumption [25].

$$J(k) = min \sum_{k=0}^{N} L[x(k), u(k)] \tag{17}$$

$$L(x(k), u(k)) = m_f(k) + \frac{s \cdot Pe(k)}{H_l} \tag{18}$$

where $m_f(k)$ is the amount of fuel consumed by the engine output torque at the $k$th moment (g), $H_l$ is the calorific value of gasoline (J/kg), $s$ is the oil–electricity equivalent factor, and $Pe(k)$ is the battery discharge/charge power.

Since the conversion efficiency of electrical energy and mechanical energy is higher than that of fuel thermal energy and mechanical energy, the concept of the oil–electricity equivalent factor $s$ is introduced to convert the consumed electrical energy into the amount of fuel consumed and then compare the fuel consumption of hybrid vehicles [26]. The value of the oil–electricity equivalent factor is closely related to the driving conditions of the vehicle and the battery SOC. For different required torques or different initial and final values of the battery's state of charge, the value of $s$ will not necessarily be the same [27].

The entire journey is divided into $N$ stages according to the time step, and the following state transition equation is established in the discrete state space according to the time sequence of the cyclic operating conditions [28]:

$$x(k+1) = f(x(k), u(k)), \ k = 0, 1, 2, \cdots, N-1 \tag{19}$$

Then, Equation (18) is substituted to calculate the cost objective function $L$ of the next stage. Under the cyclic condition, one stage needs to make a decision to enter the next stage, that is, determine the decision variable $u(k)$. Judgments are made at each stage through decision variables, and the optimal path for the state variables to pass to the next stage is selected. In this way, the optimal path for the entire working condition selected by the rules in the entire cycle is which minimizes the objective function of the cumulative cost of the system. This shortest path is described by the control variable sequence $U = \{u(0), u(1), u(2), \cdots, u(N-1)\}$.

### 3.3. Control Process Command

The state variables, decision variables, and cost objective functions in the dynamic programming problem of hybrid vehicle energy management are defined. The energy management strategy software based on the dynamic programming algorithm was developed based on the MATLAB software platform. A top-down design concept and backward development are adopted. The method deduces the actual working condition of the components from the target vehicle speed and the target acceleration of the cycle condition.

Considering the transient characteristics of the vehicle and the influence of the number of calculations in the algorithm, the sampling time of the algorithm is set to 1 s. The CCBC

condition is used to discretize the system's state variables into a grid according to the time series of the cycle conditions, and the whole cycle condition is discretized into N. Then, according to the constraints of the vehicle power system and the kinematics model of the vehicle, the feasible range of the system variable is calculated. According to the demand situation of the driving cycle, the vehicle model calculates each stage in the grid point within the feasible range. Finally, the dynamic programming algorithm is used to recursively calculate the energy distribution strategy of the state variables to minimize the total fuel consumption.

The solution process of the dynamic programming algorithm is applied step by step according to the recursive relationship provided by the state transition equation of the optimization object. The flow chart of the dynamic programming algorithm is shown in Figure 9.

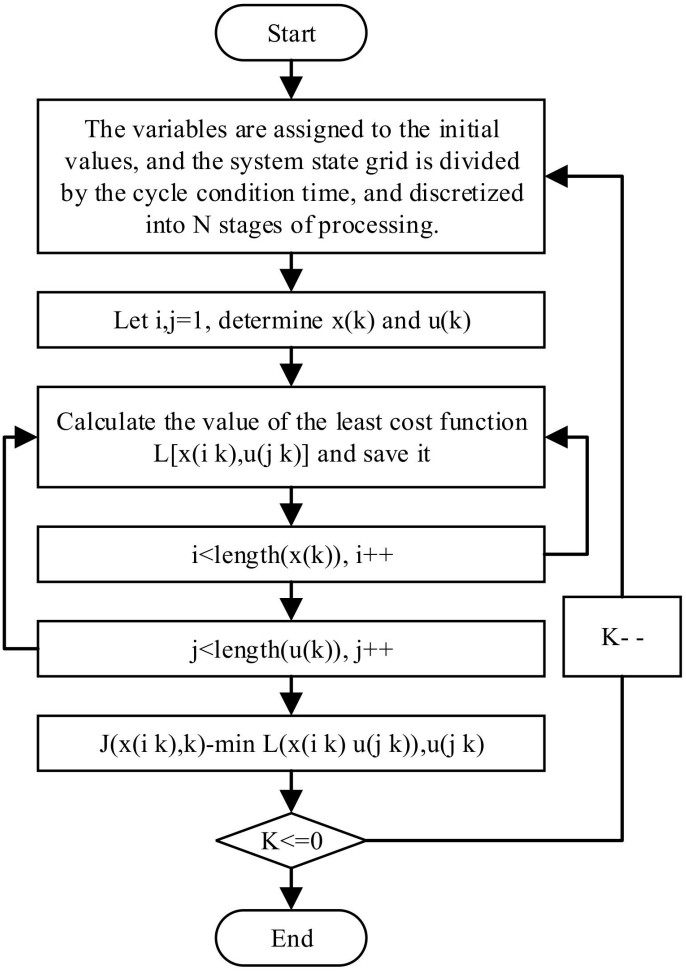

**Figure 9.** Flow chart of the dynamic programming algorithm.

*3.4. Dynamic Programming Algorithm Results Analysis*

Figure 10 shows the comparison of the demand torque and the real torque of a hybrid vehicle under the CCBC condition. It can be seen from Figure 10 that in the acceleration or uniform speed phase when the required torque is positive, the hybrid electric vehicle has good tracking performance and rapid response to the forward torque demand under the cycle working condition. At each sampling time point, the condition is basically satisfied. In the braking phase where the torque demand is negative, due to the characteristics of the hybrid vehicle, the motor transfers in the brake feedback energy to the power battery for storage, and the mechanical brake system of the motor and the brake pad of the automobile work together to slow the vehicle down. The red solid line in the figure represents the braking torque of the motor, and the difference between the black star point and the

solid red line represents the braking provided by the mechanical brake system. It can be seen that the hybrid vehicle is sensitive to the braking torque demand, and there is no misjudgment of the torque. From the overall performance point of view, the dynamic planning control strategy for the cyclic operating conditions is accurate and sensitive. Under CCBC conditions, all torque demand are basically met.

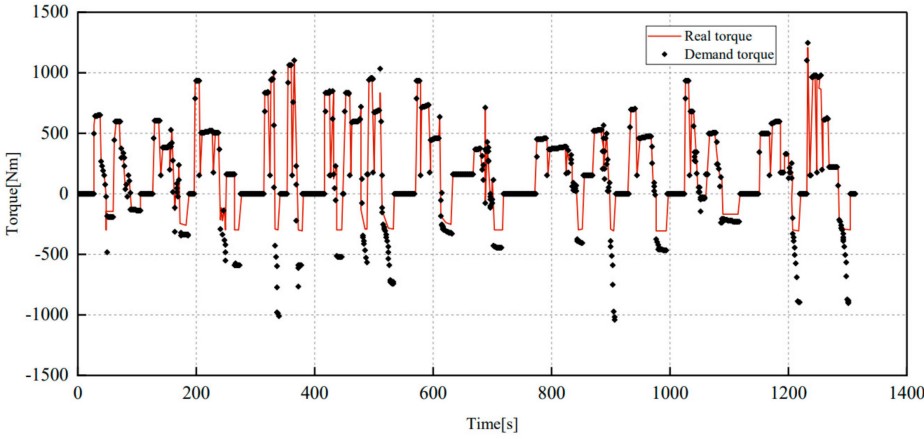

**Figure 10.** Comparison of demand torque and real torque.

Figure 11 indicates the relationship between the state of charge of the battery and the output current of the battery. This surface plot depicts that the effects of different vehicle speed on the state of charge and the output current of the battery can also be seen. From the above, the dynamic programming algorithm sets the battery to the power maintenance state, and the initial SOC is set to 0.3. It can be seen from Figure 11 that the state of charge of the battery is stable at around 0.3 in the case of a constant cycling speed throughout the cycle. The maximum variation is less than 2%, and the stability is in the power maintenance phase, which is in accordance with the SOC of the cumulative cost objective function to obtain the monotonic minimum. At the same time, the SOC of the battery is less affected by the fluctuation in the output current of the battery, which ensures that the driver can accurately identify the remaining power in the battery without being disturbed by the driving condition, and indicates the control stability of the system based on the dynamic programming algorithm.

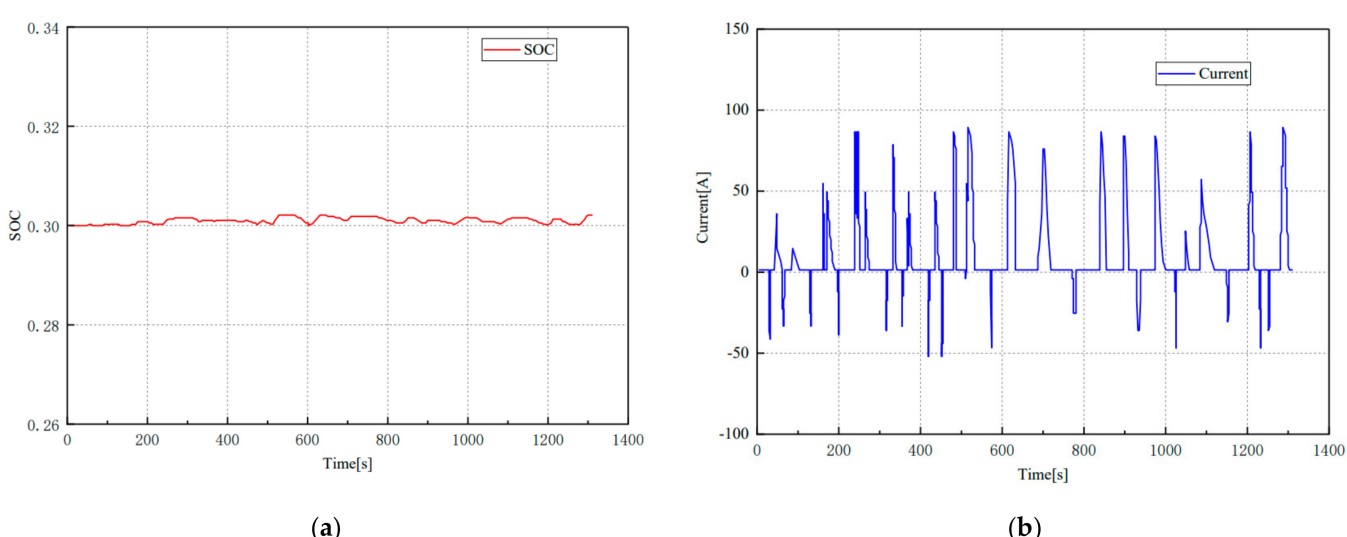

**Figure 11.** Battery SOC and battery output current as a function of time. (**a**) Battery SOC. (**b**) Battery output current.

Figure 12 shows a comparison of the torque and the speed between the motor and engine under the cycle condition. From the motor and engine speed curve in Figure 12b, it can be seen that when the engine is involved in providing the driving force, the speed is mostly maintained between 1500 r/min and 2400 r/min, and the engine maintains a certain speed (idle speed) when the required vehicle speed is 0 r/min, which is mainly due to the short downtime and optimized exhaust emission considerations that limit the control strategy of the engine. As the speed demand increases, the engine gradually participates in providing drive torque to meet the acceleration requirements of the vehicle, which is the global optimal solution of the dynamic programming algorithm. The torque distribution according to the engine and motor torque are given in the Figure 12(a) can meet the torque demand of the working condition and the lowest fuel consumption under this working condition is obtained.

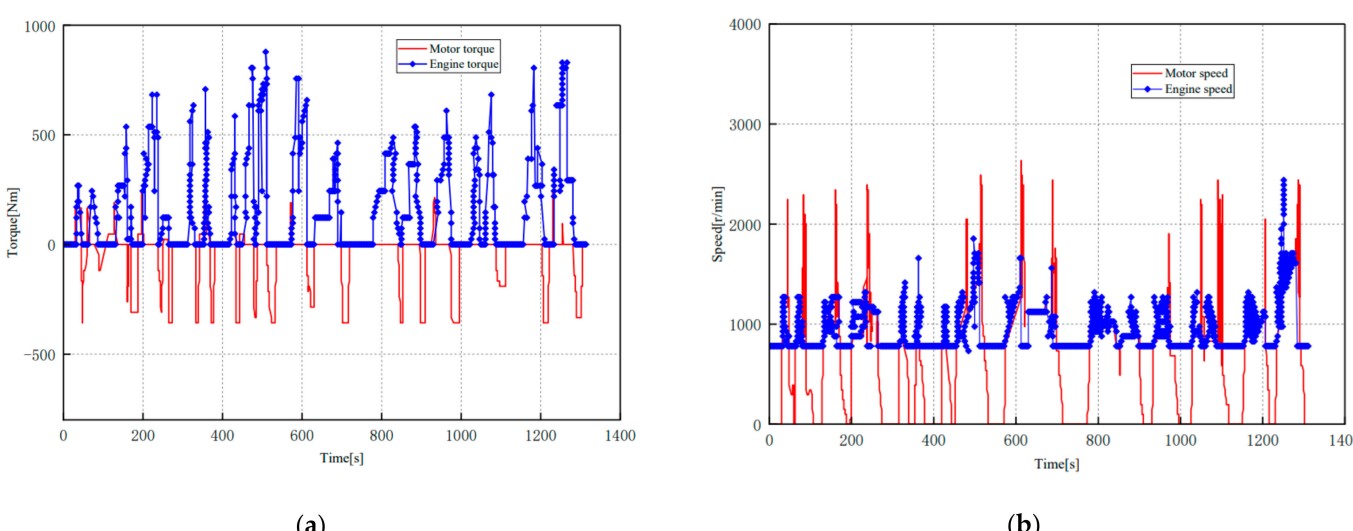

(**a**)  (**b**)

**Figure 12.** Comparison of torque and speed between engine and motor. (**a**) Engine and motor torque. (**b**) Engine and motor speed.

Figure 13 depicts the fuel consumption rate of the engine at each moment under CCBC condition. Under each working condition, the fuel consumption rate of the parallel hybrid vehicle based on the dynamic programming algorithm under CCBC condition is calculated to be 27.2 L/100 km as the Figure 13 shows.

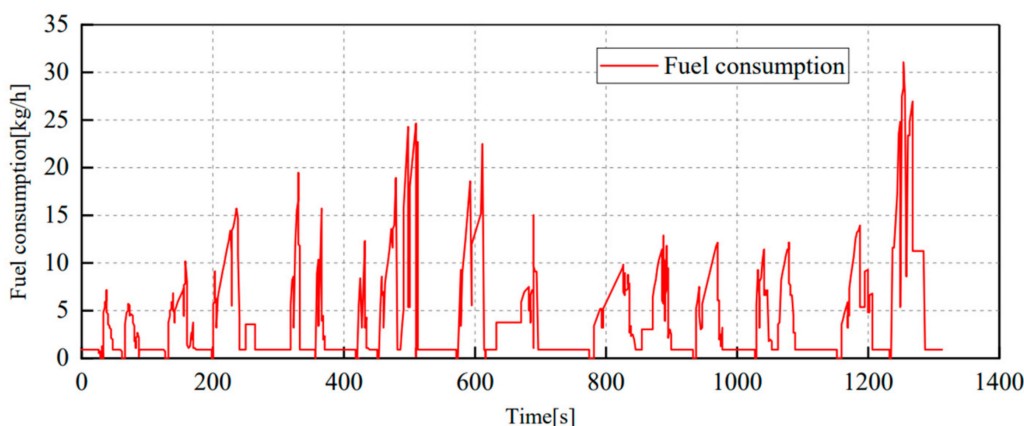

**Figure 13.** Engine fuel consumption rate.

### 3.5. Strategy Extraction Based on Dynamic Programming Algorithm Results

Since the dynamic programming algorithm needs to know in advance the demand of the driving condition for the vehicle speed, the calculation result of the algorithm is

directly related to the selected driving condition, and the control strategy under one cycle condition may not be compatible with other driving conditions. The processor must be able to quickly calculate the working condition to make control decisions in order to achieve the real-time control of the hybrid vehicle to ensure the safety of the driver. Due to the above two considerations, the dynamic programming algorithm cannot be applied online to real-world driving conditions. However, the results obtained by the dynamic programming algorithm can be used to analyze and extract valuable control rules to guide the optimization of energy management control strategies [29].

Figure 14 is a diagram of the engine operating point distribution calculated by the dynamic programming algorithm. In Figure 14, the colored gradient line represents the engine's iso-fuel consumption curve, and the green curve represents the engine's fuel consumption economic curve. Besides, the red dots represent the actual engine operating points.

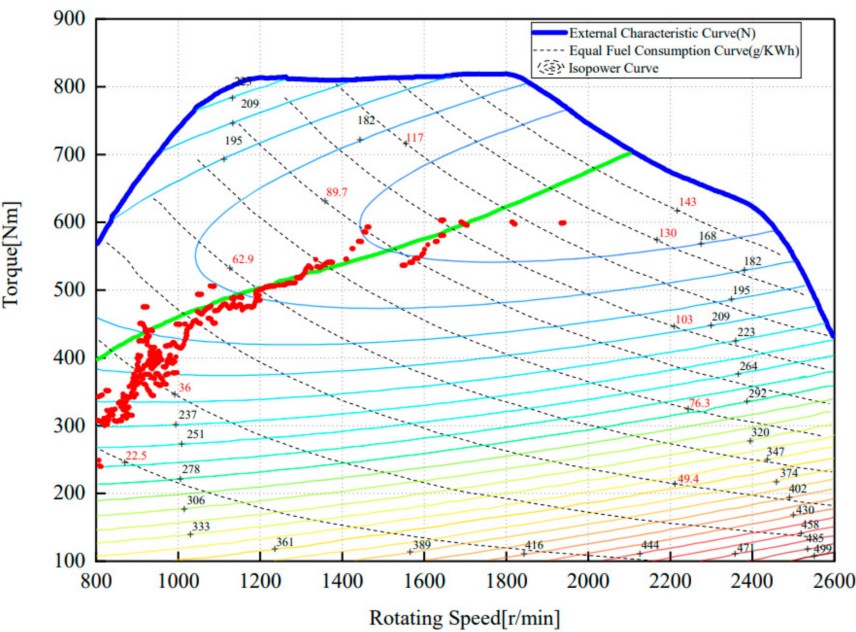

**Figure 14.** Working Point Distribution Map of the Engine Based on Dynamic Programming.

It can be seen Figure 14 that most of the engine operating points under the dynamic programming algorithm are concentrated around the fuel consumption economic curve. Due to the speed requirements of the cycle condition and the setting of the transmission system parameters, most of the working points are concentrated in the 800–1600 r/min range. In the low load region, there are a large number of operating points in the low speed and low torque region, and a small number of points are distributed in the high-speed region. The dynamic programming algorithm makes full use of the torque of the engine, so that most of the operating points of the engine are distributed in the high efficiency area, which has achieved the purpose of saving fuel.

Figure 15 illustrates the relationship between the gearbox position and the clutch engagement and disengagement under the CCBC condition. The control strategy based on the dynamic programming algorithm sets the second gear of the gearbox as the starting gear, and the sixth gear is the highest gear of the gearbox. When the required speed is 0 km/h, it still maintains the second gear, and the speed demand increases. As the gear increases, it can be seen that the speed of the cycle changes and the control strategy makes full use of the gear ratio of the gearbox, which quickly reaches a higher gear or the highest gear in the acceleration phase to use the driving torque to meet the speed demand while achieving the goal of fuel economy. On the other hand, for the algorithm adopts the control strategy of clutch-less shifting, the AMT controller will check the input and output shaft

speeds of the AMT before the forward operation, and the gear shift operation is allowed only when the ratio of the input shaft speed and the output shaft speed is close to the actual gear speed ratio. Adopting the clutch-less shifting strategy significantly reduces the separation and combined operation of the clutch, while reducing power losses and shift shocks. It can be concluded that optimal control strategy helps to reduce the shift time and improves the shift quality. The above two aspects are of great significance to improving the transmission shift strategy. According to the shift strategy proposed by this article, the optimal economy can be obtained.

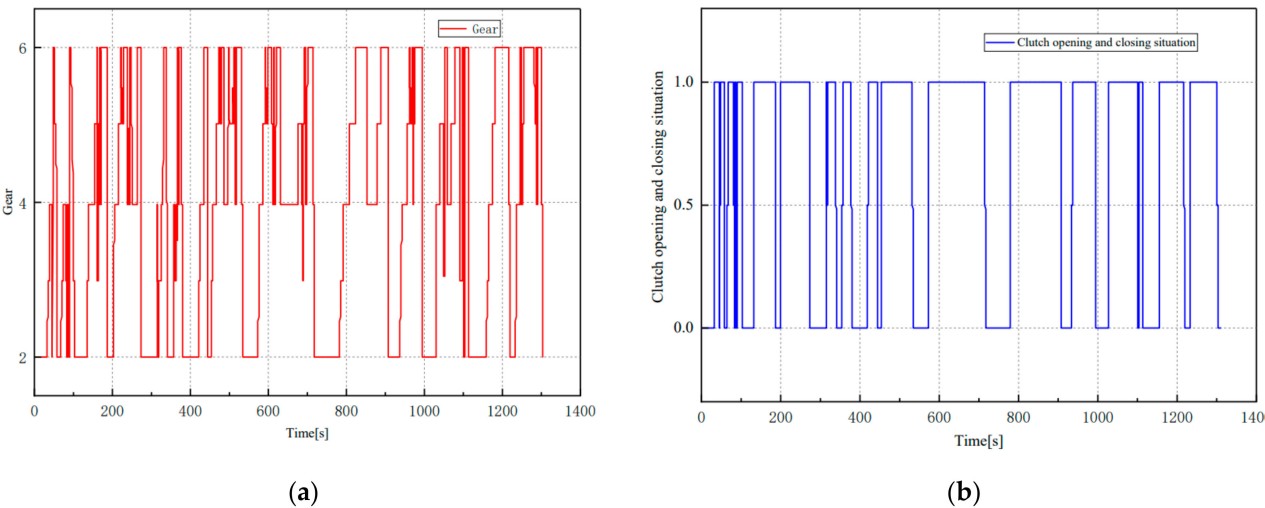

(**a**)                                                                        (**b**)

**Figure 15.** Relationship between the clutch's position and the gear with time. (**a**) Gear. (**b**) Clutch's position.

## 4. Conclusions

In this paper, the parallel plug-in hybrid vehicle is taken as the research object, and the model-based development process is adopted to realize the development of and research on a hybrid power system control strategy.

In the MATLAB/Simulink environment, the key component modeling and the controller modeling of the hybrid electric vehicle are established. Based on the operating modes of the hybrid vehicle, the control strategy is designed. An energy management strategy based on the DP algorithm is studied, and the optimal control strategy based on MATLAB software is developed, while the torque distribution rules and shift rules are analyzed. Ultimately, the optimal control strategy is used to extract the control strategy.

The optimized engine operating points are concentrated near the fuel economy curve. The optimized clutch shift strategy can the shift time and improves the shift quality. The calculation results show that DP-based energy management strategy can reasonably distribute the torque between the motor and the engine, effectively improve the fuel economy of plug-in hybrid electric vehicles, and provide guidance on the control strategy.

**Author Contributions:** Conceptualization, Y.T. and J.L.; methodology, K.L.; software, Q.Y.; validation, Q.Y., J.L. and K.L.; formal analysis, J.L.; investigation, Y.T.; resources, Y.T.; data curation, J.L.; writing—original draft preparation, J.L.; writing—review and editing, K.L.; visualization, Y.T.; supervision, Q.Y.; project administration, K.L.; funding acquisition, Y.T. All authors have read and agreed to the published version of the manuscript.

**Funding:** This research was funded by the National Key Research and Development Program Topic of China, grant number 2018YFB0105401.

**Conflicts of Interest:** The authors declare no conflict of interest.

## Abbreviations

| | |
|---|---|
| $V_{bat\_out}$ | Battery terminal voltage |
| $R_{bat}$ | Battery equivalent internal resistance |
| $V_{bat\_oc}$ | Open circuit voltage |
| **C** | Capacity of the electricity (A·h) |
| $\alpha$ | Engine throttle |
| $T_{eng}$ | Torque output of the engine (N·m) |
| $\overset{\bullet}{V}_{fuel}$ | Instantaneous fuel consumption |
| $T_{mot}$ | Actual output torque of the motor |
| $n_{mot}$ | Motor speed (r/min) |
| $T_{out}$ | Gearbox output torque (N·m) |
| $n_{in}$ | Gearbox input shaft speed (r/min) |
| F | Driving force |
| $s$ | Oil-electricity equivalent factor |

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
