# Peer review of "Optimal Control Strategy for Parallel Plug-in Hybrid Electric Vehicles Based on Dynamic Programming"

_wevj, doi:10.3390/wevj12020085_

Round 1
Reviewer 1 Report
A good paper application on optimal control is given. Problems to be shown:
a) The paper treats optimal control for parallel plug-in hybrid electric vehicles. Which are the advantages and benefits compared with the non-optimal control approaches?
b) The motivation of the approach requires a special attention. Authors' very good papers in the field deserve to be included in this discussion.
c) Besides, some more relative topics have to be included in the context of linear and nonlinear control because they are successful in various applications including optimal control: Cascade control for telerobotic systems serving space medicine (IFAC Proceedings Volumes 2011); Hybrid controller design based magneto-rheological damper lookup table for quarter car suspension (International Journal of Artificial Intelligence 2020); Hybrid data-driven fuzzy active disturbance rejection control for tower crane systems (European Journal of Control 2021).
d) I did not find the optimization problem.
e) Please also better show the controller and how did you compute its parameters.
f) The reviewer would like to suggest the authors to compare, if possible, their results with some similar optimal controller.
g) The example for application needs to be extended and better discussed in the theoretical context presented prior to that.
Author Response
Response to Reviewer 1 Comments
Point 1: The paper treats optimal control for parallel plug-in hybrid electric vehicles. Which are the advantages and benefits compared with the non-optimal control approaches?

Response 1: According to the comments, we summarize the pros and cons of each type of method at the end of paragraphs 3-6 of the introduction. Moreover, the advantages of dynamic programming methods and the value of optimal energy management strategies based on dynamic programming are added in paragraph 7.
Point 2: The motivation of the approach requires a special attention. Authors' very good papers in the field deserve to be included in this discussion.
Response 2: According to the comments, we have revised the first paragraph of the introduction to increase social needs.
Point 3: Besides, some more relative topics have to be included in the context of linear and nonlinear control because they are successful in various applications including optimal control: Cascade control for telerobotic systems serving space medicine (IFAC Proceedings Volumes 2011); Hybrid controller design based magneto-rheological damper lookup table for quarter car suspension (International Journal of Artificial Intelligence 2020); Hybrid data-driven fuzzy active disturbance rejection control for tower crane systems (European Journal of Control 2021).
Response 3: According to the comments, we have quoted the above documents in the second paragraph of the introduction
Point 4: I did not find the optimization problem.
Response 4: According to the comments, We have added Section 3.2 constraints, decision variables, optimization functions, etc. to introduce optimized control strategies. Added gearbox model and other model maps in section 2.2.
Point 5: Please also better show the controller and how did you compute its parameters.
Response 5: According to the comments, We added calculation process in section 3.2-3.3 and replaced the controller model picture.
Point 6: The reviewer would like to suggest the authors to compare, if possible, their results with some similar optimal controller.
Response 6: Due to the different simulation conditions of the papers in the field, the desired comparison effect cannot be achieved. This modification has not added this part of the content for the time being.
Point 7: The example for application needs to be extended and better discussed in the theoretical context presented prior to that.
Response 7: As optimization research is still in progress, there is currently no applicatin example that can be expanded.
Reviewer 2 Report
To attract more readers, at the beginning of the paper, I recommend defining the technical terms.
Author Response
Point 1: To attract more readers, at the beginning of the paper, I recommend defining the technical terms.
Response 1: According to the comment, we added this part at the beginning of the article.
Reviewer 3 Report
The paper is very interesting and well-developed. The following corrections should be made to improve the quality of the paper:
1. The abstract must be modified, adding the most important things found in the investigation, including the most interesting results and conclusions.
2. The problem statement should be improved, showing the weaknesses of previous research and showing the benefits of this research.
3. The introduction must be in plain text, the subsections must be removed: 1.1. Research Motivation, 1.2. Literature Review and 1.3. Main Contributions, there should be no subsections within the introduction.
4. Figure 1 should be explained in more detail.
5. There is no reference to the papers from which they obtained the mathematical equations.
6. The controller design needs to be explained further “4. Control Strategy Design ”, Figure 3 and finally do the same with section 5.“ Research on Control Strategy Based on Dynamic Programming Algorithm ”.
7. The conclusions must be improved, which must be consistent with the results obtained.
Author Response
Response to Reviewer 3 Comments
Point 1: The abstract must be modified, adding the most important things found in the investigation, including the most interesting results and conclusions.
Response 1: We adjusted the summary content based on this comment. Since the optimization goal is fuel economy, the optimization result of fuel consumption rate is added to the summary to evaluate the optimization effect of the control algorithm
Point 2: The problem statement should be improved, showing the weaknesses of previous research and showing the benefits of this research.
Response 2: First, we summarize the weaknesses of each type of method at the end of paragraphs 3-6 of the introduction. Secondly, the advantages of dynamic programming methods and the value of optimal energy management strategies based on dynamic programming are added in paragraph 7.
Point 3: The introduction must be in plain text, the subsections must be removed: 1.1. Research Motivation, 1.2. Literature Review and 1.3. Main Contributions, there should be no subsections within the introduction.
Response 3: According to the comments, the subtitle has been deleted.
Point 4: Figure 1 should be explained in more detail
Response 4: According to the comment, we adjusted the text and added an overview of the modeling work according to Figure 1.
Point 5: There is no reference to the papers from which they obtained the mathematical equations.
Response 5: According to the comments, we added references 27-30 to references in section 4.2
Point 6: The controller design needs to be explained further “4. Control Strategy Design ”, Figure 3 and finally do the same with section 5.“ Research on Control Strategy Based on Dynamic Programming Algorithm ”.
Response 6: "4. Control Strategy Design", changed to "Controller model".
The controller model picture has been changed.
"Research on Control Strategy Based on Dynamic Programming Algorithm" focuses on global optimization algorithms.
Point 7: The conclusions must be improved, which must be consistent with the results obtained.
Response 7: We added more formulas to introduce the optimal control strategy in the third section, and added two pictures to the results to better interpret and analyze the results. In the conclusion, the work of the paper is summarized first, and then the effect and significance of the optimization strategy are added.
Round 2
Reviewer 1 Report
The paper is further improved and deserves once more its acceptance.